# Dimethyl Itaconate Inhibits Melanogenesis in B16F10 Cells

**DOI:** 10.3390/antiox12030692

**Published:** 2023-03-10

**Authors:** Bo-Yeong Yu, Hoang Hai Ngo, Won Jun Choi, Young-Sam Keum

**Affiliations:** College of Pharmacy and Integrated Research, Institute for Drug Development, Dongguk University, 32 Dongguk-ro, Goyang 10326, Gyeonggi-do, Republic of Korea

**Keywords:** dimethyl itaconate (DMI), melanocortin 1 receptor (MC1R), microphthalmia-associated transcription factor (MITF), KELCH-like ECH-associated protein 1 (KEAP1), NF-E2-related factor 2 (NRF2)

## Abstract

Itaconate is a metabolite produced to counteract and resolve pro-inflammatory responses when macrophages are challenged with intracellular or extracellular stimuli. In the present study, we have observed that dimethyl itaconate (DMI) inhibits melanogenesis in B16F10 cells. DMI inhibits microphthalmia-associated transcription factor (MITF) and downregulates the expression of MITF target genes, such as tyrosinase (TYR), tyrosinase-related protein 1 (TRP-1), and tyrosinase-related protein 2 (TRP-2). DMI also decreases the level of melanocortin 1 receptor (MC1R) and the production of α-melanocyte stimulating hormone (α-MSH), resulting in the inhibition of extracellular signal-regulated kinase 1/2 (ERK1/2) and MITF activities. The structure–activity relationship (SAR) study illustrates that the α,β-unsaturated carbonyl moiety in DMI, a moiety required to target KELCH-like ECH-associated protein 1 (KEAP1) to activate NF-E2-related factor 2 (NRF2), is necessary to inhibit melanogenesis and knocking down Nrf2 attenuates the inhibition of melanogenesis by DMI. Together, our study reveals that the MC1R-ERK1/2-MITF axis regulated by the KEAP1-NRF2 pathway is the molecular target responsible for the inhibition of melanogenesis by DMI.

## 1. Introduction

Skin pigmentation results from the distribution of melanin in keratinocytes. Melanin is initially synthesized and stored in the melanosome, a unique organelle located in the cytoplasm of melanocytes [1]. As melanocytes undergo maturation, the melanosome is translocated and deposited in neighboring keratinocytes, resulting in the spread of melanin pigments in the epidermis [2,3]. The primary role of melanin is to protect against the mutagenic and oxidative damages caused by ultraviolet B (UVB) in the skin: melanin not only absorbs the energy generated by UVB but also acts as a powerful free radical scavenger [4]. On the other hand, it is known that excessive production of melanin is also responsible for hyperpigmentation skin disorders, including melanoma [5]. Therefore, the development of anti-melanogenic agents is considered feasible for cosmetic purposes and melanoma therapy [6]. The skin can be protected by the external use of chemical sunscreen lotions that reflect UVB radiation or physical sunscreen lotions that absorb UVB radiation [7]. In addition, many studies are undergoing to find out new whitening agents [8] which can selectively target the melanogenic pathway in the skin without disturbing homeostasis [9].

Succinate, a metabolite of the tricarboxylic acid (TCA) cycle, accumulates in macrophages during inflammation and contributes to the activation of pro-inflammatory responses: succinate oxidation by succinate dehydrogenase (SDH) induces the production of pro-inflammatory proteins and inhibits the production of anti-inflammatory proteins [10]. To counteract pro-inflammatory responses by succinate, another metabolite in the TCA cycle, cis-aconitate, is decarboxylated by the enzyme IRG1 (Immune-regulated gene 1) to produce itaconate in macrophages [11,12]. Itaconate is catabolized into itaconyl-CoA and citramalyl-CoA, the latter of which is recycled back to pyruvate and acetyl-CoA by citrate lyase subunit beta-like (CLYBL) [13]. Itaconate was initially identified as a product of citric acid distillation in 1836, and its commercial production has been carried out by Aspergillus species in vivo [14]. Due to the presence of a reactive methylene group, itaconate has been utilized as a building block for the preparation of industrial polymers and bioactive compounds in the agricultural and medicinal industries [15]. 

In 2011, an implication of itaconate in the metabolism was unexpectedly demonstrated: the production of itaconate significantly increases in the lungs of Mycobacterium tuberculosis-infected mice [16] and in the intracellular compartment of glia-like VM-M3 cells [17]. Subsequent studies have demonstrated that diverse oxidants and electrophiles can contribute to the production and accumulation of itaconate [18]. It is now established that itaconate is a metabolite critical for antimicrobial defense, anti-inflammation, and mammalian immunity [19]. The molecular targets of itaconate include not only the inhibition of succinate dehydrogenase (SDH) [20] but also the activation of ATF3 to inhibit IκBξ target genes [21] and the inhibition of KEAP1 to activate NRF2 [22]. In the present study, we have demonstrated that dimethyl itaconate (DMI), an alkylated derivative of itaconate, inhibits melanogenesis in B16F10 cells and conducted in-depth investigations to reveal the molecular mechanisms underlying how DMI inhibits melanogenesis in B16F10 cells. 

## 2. Materials and Methods

### 2.1. Cell Culture, Chemicals, Antibodies, and Plasmids 

Dulbecco’s modified eagle media (DMEM) (cat. No. CM002-050), RPMI-1640 media, fetal bovine serum (FBS) (cat. No. F0900-050), phosphate-buffered saline (PBS) (cat. No. P2101-050), and penicillin/streptomycin (Pen/Strep) (cat. No. CA005-010) were purchased from GenDEPOT (Austin, TX, USA). B16F10 cells and SK-MEL-3 cells were purchased from the Korean cell line bank (Seoul, Republic of Korea). 293T cells and HaCaT cells were purchased from the American tissue culture collection (Manassas, VA, USA). Itaconate (cat. No. 129204), dimethyl itaconate (DMI) (cat. No. 109533), N-acetyl cysteine (NAC) (cat. No. A7250), L-glutamine (cat. No. G8540), HEPES (cat. No. H4034), and sulforaphane (cat. No. S4441) were purchased from Sigma (St. Louis, MO, USA). Monomethyl itaconate (MMI) (cat. No. I0269) was purchased from TCI (Chuo City, Tokyo, Japan). B16F10 cells, 293T cells, and HaCaT cells were cultured with 1x DMEM containing 10% FBS and 1% Pen/Strep at 37 °C in a 5% CO_2_ incubator. SK-MEL-3 cells were cultured with RPMI-1640 media containing 10% FBS, L-glutamine (300 mg/L), and 25 mM HEPES at 37 °C in a 5% CO_2_ incubator. Antibodies against MC1R (cat. No. ab180776) and ERK1/2 (cat. No. ab184699) were purchased from Abcam (Cambridge, UK). Antibodies against Phospho-specific ERK1/2 at Thr202/204 (cat. No. 9101S) and NRF2 (cat. No. 12721S) were purchased from Cell Signaling Technology (Beverly, MA, USA). Antibodies against actin (cat. No. sc-8432), PAH (cat. No. sc-271258), MITF (cat. No. sc-56725), Tyrosinase (cat. No. sc-20035), TRP-1 (cat. No. sc-166857), and TRP-2 (cat. No. sc-74439), were purchased from Santa Cruz Biotechnology (Santa Cruz, CA, USA). Phospho-specific MITF antibody at Ser73 (cat. No. LS-C199259) was purchased from LSBio (Seattle, WA, USA). DNA oligonucleotides were purchased from Macrogen (Seoul, Republic of Korea).

### 2.2. Measurement of Intracellular Melanin

B16F10 cells and SK-MEL-3 cells were seeded at a density of 1 × 10^5^ cells and 3 × 10^5^ cells in 6-well culture dishes, respectively. After treatment, B16F10 cells and SK-MEL-3 cells were washed with 1x PBS and dissolved in 100 μL of 1N NaOH. The level of intracellular melanin was measured by the spectrophotometer at the wavelength of 405 nm. 

### 2.3. Trypan Blue Exclusion Assay

B16F10 cells and SK-MEL-3 cells were seeded at a density of 1 × 10^5^ cells in 24-well culture plates (around 70% confluent). After treatment, cells were trypsinized and stained with 0.2% trypan blue solution. The number of viable cells was counted using a hemocytometer under the microscope.

### 2.4. RNA Extraction and Real-Time Reverse Transcription-Polymerase Chain Reaction (RT-PCR)

Total RNA was extracted using a Hybrid-R RNA extraction kit (GeneAll, Seoul, Republic of Korea). Synthesis of cDNA from total RNA (1 μg) was conducted using AmfiRivert cDNA Synthesis Platinum master mix (GenDEPOT, Austin, TX, USA). Real-time RT-PCR analysis was performed using SYBR mix (ELPIS Biotech, Daejeon, Republic of Korea) on the CFX384 real-time system as recommended by the manufacturer (BioRad, Hercules, CA, USA), and the mRNA level of individual genes was normalized by that of GAPDH. PCR primer sequences against individual genes are listed (Table 1). 

### 2.5. Western Blot Analysis

After treatment, cell pellets were collected, resuspended with 1x RIPA lysis buffer [50 mM Tris-HCl at pH = 8.0, 150 mM NaCl, 1% NP-40, 0.5% deoxycholic acid, 0.1% sodium dodecyl sulfate (SDS), 1 mM Na_3_VO_4_, 1 mM dithiothreitol (DTT), 1 Mm phenylmethyl sulfonyl fluoride (PMSF)], and incubated on ice for 1 h. Cell lysates were collected by centrifugation at 13,000 rpm for 15 min, and protein concentration was measured using a BCA protein assay kit (TAKARA, Seoul, Republic of Korea). An equal amount of cell lysates was resolved by SDS-PAGE and transferred to PVDF membranes (Merck-Millipore Korea, Daejeon, Republic of Korea). The membranes were incubated in blocking buffer (5% skim milk in 1x PBS with 0.1% Tween-20, 1x PBST) for 1 h and hybridized with appropriated primary antibodies in 1x PBS overnight at 4 °C. After washing three times with 1x PBST for 30 min, the membranes were hybridized with horseradish peroxidase (HRP)-conjugated secondary antibodies (Thermo-Fischer Scientific, Waltham, MA, USA) for 1 h at 4 °C and washed three times with 1x PBST for 30 min. The membranes were visualized by using enhanced chemiluminescence (ECL) detection system. Western blot was conducted in triplicates, and the images of the films were subjected to densitometry analysis. 

### 2.6. Generation of Stable Cells by Lentiviral Transduction

The pGreenFire-GFP-luciferase lentiviral expression vector was purchased from Systems Biosciences (Palo Alto, CA, USA), and lentiviral helper plasmids (pMD2.G and psPAX2) were acquired from Addgene (Cambridge, MA, USA). The construction of pGreenFire-M-box/E-box-GFP-luciferase plasmid (Figure 3A) and pGreenFire-ARE-GFP luciferase plasmid [23] is described. To perform lentiviral transduction in B16F10 cells, 293T cells were transfected with 1 μg of pGreenFire-M-box/E-box-GFP-luciferase plasmid and pGreenFire-ARE-GFP-luciferase plasmid together with 1 μg of lentiviral helper plasmids (psPAX2 and pMD2.G). At 72 h post-transfection, the viral supernatant was collected, filtered, and transduced into B16F10 cells in the presence of 5 µg/mL polybrene (Merck-Millipore Korea, Daejeon, Republic of Korea). Transduced B16F10 cells were selected with puromycin (Invivogen, San Diego, CA, USA) at the concentration of 2 μg/mL for 48 h. 

### 2.7. Firefly Luciferase Assay

B16F10-M-box/E-box-GFP-luciferase cells and B16F10-ARE-GFP-luciferase cells were seeded at a density of 1 × 10^5^ cells in 24-well culture plates. After treatment, B16F10-M-box/E-box-GFP-luciferase cells and B16F10-ARE-GFP-luciferase cells were washed with 1x PBS and lyzed with luciferase lysis buffer [0.1 M potassium phosphate buffer at pH 7.8, 1% Triton X-100, 1 mM DTT, 2 mM EDTA] for 1 h. The resulting firefly luciferase activity was monitored by GLOMAX Multi-system (Promega, Madison, WI, USA), and the luciferase activity was normalized by protein concentration of lysates.

### 2.8. Measurement of α-Melanocyte Stimulating Hormone (α-MSH)

B16F10 cells were seeded at a density of 3 × 10^5^ cells in 60 mm culture plates, and the amount of α-MSH produced by B16F10 cells was measured by mouse α-MSH ELISA kit (LSBio, Seattle, WA, USA) according to the manufacturer’s protocol.

### 2.9. Knocking Down Mitf and Nrf2 mRNAs by Lentiviral Transduction

Knocking down Mitf and Nrf2 mRNAs in B16F10 cells was conducted by lentiviral transduction using pLKO.1-puro lentiviral vector (Addgene, Cambridge, MA, USA). After double digestion of pLKO.1-puro lentiviral vector with EcoRI and AgeI, DNA nucleotides targeting Mitf and Nrf2 genes (Table 2) were ligated to pLKO.1 vector. 1 μg of pLKO1 vectors ligated with DNA nucleotides targeting Mitf (pLKO1-shMitf) and Nrf2 (pLKO1-shNrf2) genes were co-transfected with 1 μg of lentiviral helper plasmids (pMD2.G and psPAX2) in 293T cells. At 72 h post-transfection, the viral supernatant was collected and transduced to B16F10 cells with 5 µg/mL polybrene (Merck-Millipore Korea, Daejeon, Republic of Korea). Transduced B16F10 cells were selected with puromycin (2 μg/mL) for 48 h.

### 2.10. Statistics

Statistical analysis was conducted using Student’s *t*-test. Asterisks indicate statistical significances of * *p* < 0.05, ** *p* < 0.01, and *** *p* < 0.001. 

## 3. Results

### 3.1. Dimethyl Itaconate (DMI) Inhibits the Production of Melanin in B16F10 and SK-MEL-3 Cells

In order to examine whether itaconate possesses an inhibitory effect on the production of melanin, we exposed itaconate, monomethyl itaconate (MMI), and dimethyl itaconate (DMI) (Figure 1A) to murine melanoma B16F10 cells, and compared their effects on the production of melanin: we speculated that methylated itaconate would translocate the plasma membrane better than itaconate due to an increase in hydrophobic property [24]. Our results demonstrate that DMI suppressed the production of melanin stronger than MMI, but itaconate failed to do so in B16F10 cells (Figure 1B). We made an analogous observation when we exposed itaconate, MMI, and DMI to human melanoma SK-MEL-3 cells (Figure 1C). DMI did not affect the viability of B16F10 cells and SK-MEL-3 cells as measured by trypan blue exclusion assay (Figure 1D). These results indicate that DMI inhibits the production of melanin without exhibiting cytotoxicity. 

### 3.2. DMI Inhibits TYR, TRP-1, and TRP-2 in B16F10 Cells

Melanogenesis requires tyrosine as a substrate, which undergoes a series of enzymatic transformations by tyrosinase (TYR), tyrosinase-related protein 1 (TRP-1), and tyrosinase-related 2 (TRP-2) [25], resulting in the production of eumelanin and pheomelanin (Figure 2A) [26]. Because DMI inhibited the production of melanin in B16F10 cells (Figure 1B), we next examined whether DMI could affect the level of TYR, TRP-1, and TRP-2 in B16F10 cells. Our results show that DMI suppressed the expression of TYR, TRP-1, and TRP-2 in B16F10 cells (Figure 2B), and this event was associated with transcriptional inhibition of Tyr, Trp-1, and Trp-2 (Figure 2C). Together, these results indicate that the inhibition of melanogenesis by DMI can be ascribed to the inhibition of TYR, TRP-1, and TRP-2 in B16F10 cells. 

### 3.3. DMI Inhibits MITF in B16F10 Cells

Microphthalmia-associated transcription factor (MITF) is a transcription factor that regulates the expression of TYR, TRP-1, and TRP-2 [28] by binding to the M-box (5’-TCATGTG-3’) and/or the E-box (5’-CACGTG-3’), two regulatory motifs existing in the promoter of target genes of MITF [29]. To examine whether MITF is implicated in a decrease in the level of TYR, TRP-1, and TRP-2 by DMI, we have established B16F10-M-box/E-box-GFP-luciferase cells by lentiviral transduction (Figure 3A) and exposed them to DMI. Our result shows that DMI significantly inhibited the expression of MITF-dependent luciferase activity in B16F10-M-box/E-Box-GFP-luciferase cells (Figure 3B). In addition, DMI downregulated the expression of MITF in B16F10 cells (Figure 3C), and this event occurred at the transcriptional level (Figure 3D). Together, these results illustrate that transcriptional inhibition of MITF by DMI can be attributable to the inhibition of TYR, TRP-1, and TRP-2 in B16F10 cells. 

### 3.4. DMI Inhibits the α-MSH/MC1R-ERK1/2-MITF Axis in B16F10 Cells 

MITF is regulated by three membrane receptors, e.g., the low-density lipoprotein receptor-related proteins 5/6 (LRP5/6), the receptor tyrosine kinase c-Kit, and the melanocortin 1 receptor (MC1R) [4,30,31]. When activated, the LRP5/6, the c-Kit, and the MC1R transmit signals by activating downstream kinase pathways, resulting in MITF phosphorylation activation at Ser73 [32] (Figure 4A). To address which receptor-dependent signaling cascade(s) are implicated in the inhibition of MITF by DMI, we exposed B16F10 cells to DMI and conducted real-time RT-PCR against the LPR5, the c-Kit, and the MC1R. Our results show that DMI selectively inhibited the level of the Mc1r but not that of the Lrp5 and the c-Kit mRNAs in B16F10 cells (Figure 4B). We observed that DMI also decreased the production of α-melanocyte stimulating hormone (α-MSH), a ligand of MC1R, in B16F10 cells (Figure 4C). In addition, DMI suppressed the activity of downstream signaling cascades of the α-MSH/MC1R, e.g., ERK1/2 phosphorylation at Thr202/204 and MITF phosphorylation at Ser73 in B16F10 cells (Figure 4D). Together, these results illustrate that DMI inhibits melanogenesis in B16F10 cells by targeting the α-MSH/MC1R-ERK1/2-MITF axis.

### 3.5. The α,β-Unsaturated Carbonyl Moiety in DMI Is Required to Inhibit Melanogenesis in B16F10 Cells 

Since DMI contains the α,β-unsaturated carbonyl moiety, we next examined whether this moiety is necessary for the inhibition of melanogenesis by DMI. To address this issue, we have synthesized two derivatives of DMI lacking the α,β-unsaturated carbonyl moiety (Appendix A): one whose double bond is saturated (Derivative 1, dimethyl 2-methylsuccinate) and the other whose carbonyl groups are completely removed (Derivative 2, 4-methoxy-2-(methoxymethyl)but-1-ene) (Figure 5A). Then, we exposed B16F10 cells to DMI and these derivatives and examined their effects on melanogenesis. Our results show that, unlike DMI, derivatives 1 and 2 failed to inhibit the production of melanin in B16F10 cells (Figure 5B), MITF-dependent luciferase activity in B16F10-M-Box/E-Box-luciferase cells (Figure 5C) and the level of Mitf, Tyr, Trp-1, and Trp-2 mRNAs in B16F10 cells (Figure 5D). Together these data illustrate that the α,β-unsaturated carbonyl moiety in DMI is required to suppress melanogenesis in B16F10 cells. 

### 3.6. NRF2 Is Required for the Inhibition of Melanogenesis by DMI in B16F10 Cells

It is known that oxidative stress is closely associated with melanogenesis, while the detailed molecular mechanisms are largely unclear. Inhibition of oxidative stress in cells can occur by treatment of direct antioxidants or by transcriptional induction of phase II antioxidant enzymes via NRF2 activation. While DMI can act as a direct antioxidant is currently unclear at present, it is known that itaconate can exert antioxidant effects by targeting KEAP1 to activate NRF2: the α,β-unsaturated carbonyl moiety in itaconate acts as a Michael acceptor for KEAP1 [22], leading us to speculate that NRF2 activation by DMI might be associated with the inhibition of melanogenesis. Consistent with this notion, we observed that DMI increased the expression of NRF2 (Figure 6A) and the level of Nrf2 mRNA (Figure 6B) in B16F10 cells. Together with itaconate and MMI, DMI also activated NRF2-dependent antioxidant response element (ARE)-luciferase activity in B16F10-ARE-GFP-luciferase cells (Figure 6C): we established B16F10-ARE-luciferase cells in an analogous way with B16F10-M-Box/E-Box-GFP-luciferase activity [23]. We also noticed that the inhibition of melanin production (Figure 6D), MITF expression (Figure 6E), and MITF target gene transcription (Figure 6F) by DMI did not occur when Nrf2 was silenced in B16F10 cells (Appendix A). Together, these results illustrate that the inhibition of melanogenesis by DMI requires NRF2 in B16F10 cells. 

## 4. Discussion

The tricarboxylic acid (TCA) cycle involving a series of enzyme-catalyzed reactions for ATP production exists at the core of cellular metabolism and provides essential metabolic intermediates required for the regulation of various biological reactions [33,34]. In the present study, we have observed that DMI, but not itaconate, significantly inhibited the production of melanin in B16F10 cells (Figure 1B) and SK-MEL-3 cells (Figure 1C), suggesting that modification of itaconate in a way to promote its hydrophobic property could increase the anti-melanogenic effects of itaconate. Like DMI, 4-octyl itaconate is another derivative that is widely used for the examination of the activity of itaconate [35,36], but we are unaware whether 4-octyl itaconate could exhibit similar anti-melanogenic effects in B16F10 cells and SK-MEL-3 cells. 

Previously, Shin et al. have demonstrated that adenoviral expression of NRF2 or treatment of wortmannin using as an inhibitor of the PI3K/AKT signaling pathway suppressed melanogenesis in normal human melanocytes (NHMCs) [37]. On the other hand, Jang et al. have demonstrated that treatment of DMI reduced α-MSH-induced pigmentation by increasing AKT activation phosphorylation at Ser473 in B16F10 cells [38]. While the role of the PI3K/AKT pathway in the inhibition of melanogenesis by DMI is still enigmatic, we have demonstrated that DMI inhibits MITF (Figure 3) and its melanogenic targets such as TYR, TRP-1, and TRP-2 (Figure 2) by affecting the α-MSH/MC1R-ERK1/2-MITF axis (Figure 4). We also observed that the production of melanin and MITF activation by DMI did not occur in B16F10 cells when Nrf2 was silenced (Figure 6). Together, these data illustrate that NRF2 is critically implicated in the inhibition of melanogenesis by DMI. In addition, we observed that treatment of DMI or knocking down Mitf decreased the level of phenylalanine hydroxylase (PAH) in B16F10 cells (Appendix A): phenylalanine hydroxylase (PAH) is a rate-limiting enzyme that catalyzes the conversion of phenylalanine to tyrosine [39]. While additional studies are required to address the detailed molecular mechanisms, we are tempted to speculate that the inhibition of PAH by DMI might serve as another mechanism contributing to the inhibition of melanogenesis by decreasing the supply of intracellular tyrosine, a starting material for melanogenesis (Figure 2A). 

We have observed that DMI derivatives lacking an unsaturated double bond (Derivative 1) or carbonyl group (Derivative 2) failed to inhibit melanogenesis in B16F10 cells (Figure 5), suggesting that compounds possessing the α,β-unsaturated carbonyl moiety can be potential candidates for anti-melanogenic agents. It is assumed that selected compounds bearing the α,β-unsaturated carbonyl group are effective NRF2 activators due to their high reactivity against thiols in KEAP1 [40], and many of them possess significant inhibitory effects on the inflammatory responses [41]. In the present study, we observed that DMI significantly induced NRF2, and it failed to inhibit melanogenesis when Nrf2 was silenced (Figure 6). Therefore, it is possible that creating intracellular antioxidant capacity by treatment of NRF2 activators might be able to exhibit anti-melanogenic effects, and this conjecture is supported by our observation that treatment of sulforaphane, a natural compound that targets KEAP1 to induce NRF2 [42], caused the inhibition of melanogenesis (Appendix A).

Some melanogenic enzymes consume energy and generate superoxide anion and hydrogen peroxide as by-products, creating pro-oxidant cellular states and sensitizing the epidermis to oxidative stress [43]. Therefore, the maintenance of relevant redox homeostasis in the epidermis seems important for the regulation of the melanogenic pathway. In this regard, another potential mechanism for the inhibition of melanogenesis by DMI can be attributed to replenishing reduced glutathione (GSH), creating an intracellular antioxidant milieu via NRF2 activation. This conjecture is supported by our observation that treatment of N-acetyl cysteine, a glutathione precursor, significantly inhibited melanogenesis in B16F10 cells (Appendix A). Because NRF2 is responsible for the expression of the GSH transporter in the membrane, the cysteine/glutamate transporter (solute carrier family 7 member 11, SLC7A11) and that of GSH-synthesizing enzymes (glutamate-cysteine ligase catalytic subunit, GCLC and glutamate-cysteine ligase modifier Subunit, GCLM) [44], we are tempted to speculate that transcriptional activation of SLC7A11, GCLC, and GCLM by DMI might have contributed, at least in part, to the inhibition of melanogenesis. In the same tenet, the inhibitory mechanism of melanogenesis by DMI can be extended to an increase in the amount of other ROS quenchers such as NADPH, thioredoxin, and peroxiredoxin, considering that NADPH:quinone reductase 1 (NQO1), thioredoxin reductase 1 (TXNRD1), and peroxiredoxin reductase 12 (PRDX1) are NRF2 target enzymes [45].

## 5. Conclusions

In the present study, we have identified that DMI inhibits melanogenesis in B16F10 cells. We also showed that DMI inhibits the expression of MITF target genes by targeting the MC1R-ERK1/2-MITF axis. In addition, we have demonstrated that the inhibition of melanogenesis by DMI occurs in an NRF2-dependent manner. Therefore, it will be interesting to observe whether the topical application of DMI might be effective in suppressing melanogenesis in the clinical setting.

## Figures and Tables

**Figure 1 antioxidants-12-00692-f001:**
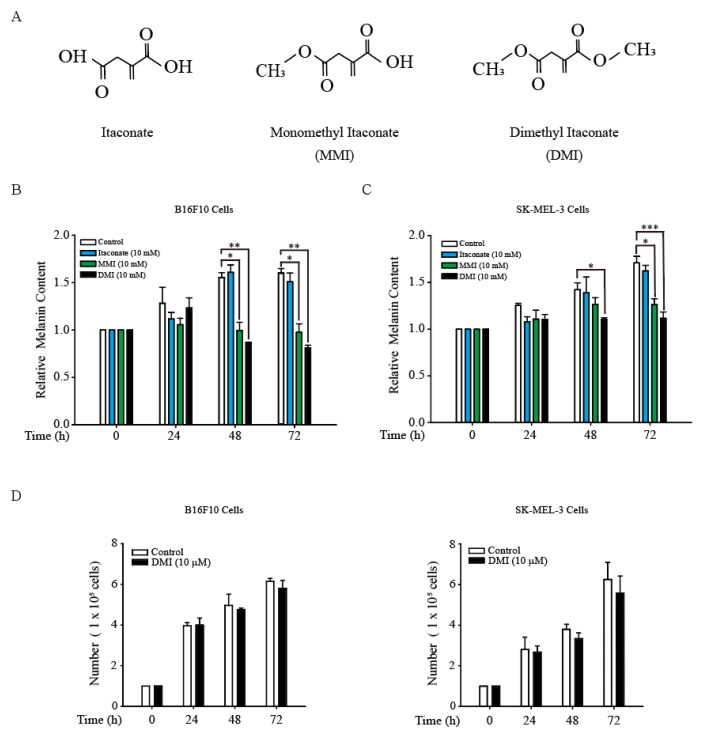
DMI inhibits the production of melanin without affecting the viability of B16F10 cells and SK-MEL-3 cells. (**A**) Chemical Structures of itaconate, monomethyl itaconate (MMI), dimethyl itaconate (DMI). (**B**) DMI inhibits the production of melanin in B16F10 cells. B16F10 cells were exposed to itaconate, MMI, and DMI, and the amount of melanin was measured (*n* = 3). (**C**) DMI inhibits the production of melanin in SK-MEL-3 cells. SK-MEL-3 cells were exposed to itaconate, MMI, and DMI, and the amount of melanin was measured (*n* = 3). (**D**) DMI does not affect the viability of B16F10 Cells and SK-MEL-3 cells. B16F10 cells (Left Panel) and SK-MEL-3 cells (Right Panel) were exposed to DMI, and the viability was measured by trypan blue exclusion assay (*n* = 3).

**Figure 2 antioxidants-12-00692-f002:**
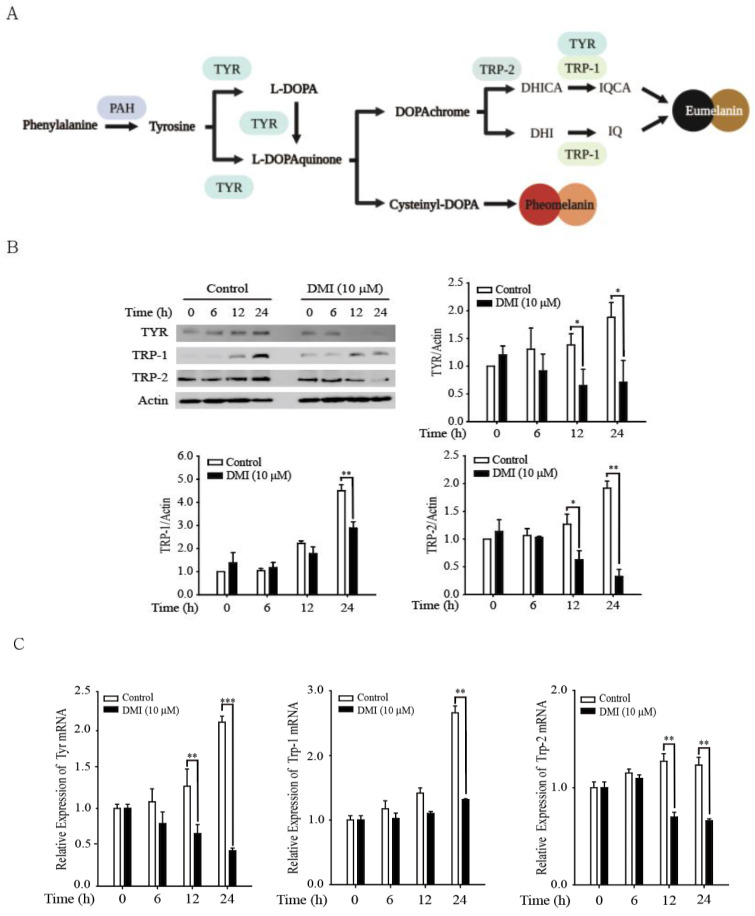
DMI downregulates the expression of TYR, TRP-1, and TRP-2 in B16F10 cells by inhibiting transcription. (**A**) The melanogenesis pathway. Melanogenesis requires tyrosine as a substrate, which is produced from phenylalanine by phenylalanine hydroxylase (PAH). Conversion of tyrosine to L-DOPA (L-3,4-dihydroxyphenylalanine) and L-DOPAquinone is catalyzed by TYR. L-DOPAquinone is further converted to DOPAchrome and cysteinyl-DOPA. While serial conversion from tyrosine into L-DOPA and L-DOPAquinone is mediated by the catecholase activity of TYR, direct conversion of tyrosine to L-DOPAquinone is also possible due to the cresolase activity of TYR [27]. Conversion of DOPAchrome to DHICA (5,6-dihydroxyindole-2-carboxylic acid) is catalyzed by TRP-2, and the conversion of DHICA to IQCA (indole-5,6-quinone-carboxylic acid) and that of DHI (5,6-dihydroxyindole) to IQ (indole-5,6-quinone) is catalyzed by TRP-1. (**B**) DMI inhibits the expression of TYR, TRP-1, and TRP-2 in B16F10 cells. B16F10 cells (1.2 × 10^6^ cells) were exposed to DMI at various times, and the protein expression of TYR, TRP-1 and TRP-2 was measured by Western blot analysis. The images of the films were analyzed by densitometry and calculated (*n* = 3). Representative images of the films are provided. (**C**) DMI inhibits the level of Tyr, Trp-1, and Trp-2 mRNAs in B16F10 cells. B16F10 cells (1.2 × 10^6^ cells) were exposed to DMI at various times, and the levels of Tyr, Trp-1, and Trp-2 mRNAs were measured by real-time RT-PCR analysis (*n* = 5).

**Figure 3 antioxidants-12-00692-f003:**
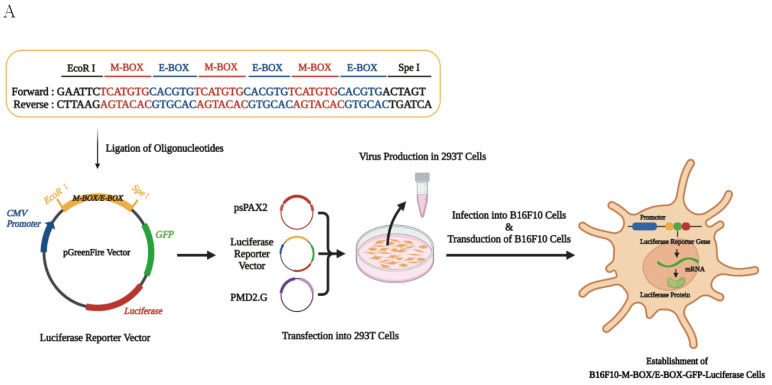
DMI inhibits MITF in B16F10 cells. (**A**) A diagram demonstrating the generation of B16F10-M-box/E-box-GFP-luciferase cells. (**B**) DMI inhibits MITF-dependent luciferase activity. Established B16F10-M-Box/E-Box-GFP-luciferase cells (1 × 10^5^ cells) were exposed to DMI for 24 h, and the luciferase activity was measured (*n* = 5). (**C**) B16F10 cells (1.2 × 10^6^ cells) were exposed to DMI at various times, and the protein expression of MITF was measured by Western blot analysis. The images of the films were analyzed by densitometry and calculated (*n* = 3). Representative images of the films are provided. (**D**) B16F10 cells (1.2 × 10^6^ cells) were exposed to DMI at various times, and the level of Mitf mRNA was measured by real-time RT-PCR analysis (*n* = 5).

**Figure 4 antioxidants-12-00692-f004:**
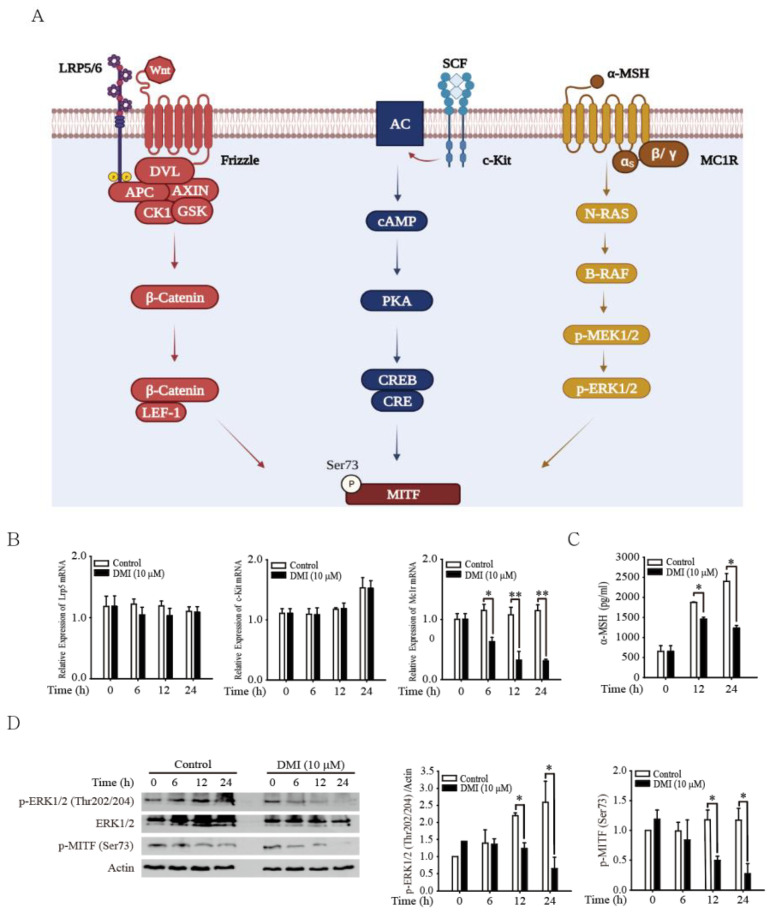
DMI inhibits melanogenesis by targeting the α-MSH/MC1R-ERK1/2-MITF axis in B16F10 cells. (**A**) A diagram demonstrating how the activation of three membrane receptors (the LRP5/6, the c-Kit, and the MC1R) elicits phosphorylation activation of MITF at Ser73. (**B**) DMI downregulates the level of Mc1r mRNA in B16F10 cells. B16F10 cells (1.2 × 10^6^ cells) were exposed to DMI, and the mRNA levels of Lrp5/6 (Left), c-Kit (Middle), and Mc1r (Right) were measured by real-time RT-PCR (*n* = 5). (**C**) DMI inhibits the production of α-MSH in B16F10 cells. B16F10 cells (3 × 10^5^ cells) were exposed to DMI, and the amount of α-MSH was measured by ELISA (*n* = 5). (**D**) DMI inhibits the phosphorylation activation of ERK1/2 and MITF in B16F10 cells. Phosphorylation of ERK1/2 at Thr202/204 and MITF at Ser73 was measured by Western blot analysis using phospho-specific antibodies. The images of the films were analyzed by densitometry and calculated (*n* = 3). Representative images of the films are provided.

**Figure 5 antioxidants-12-00692-f005:**
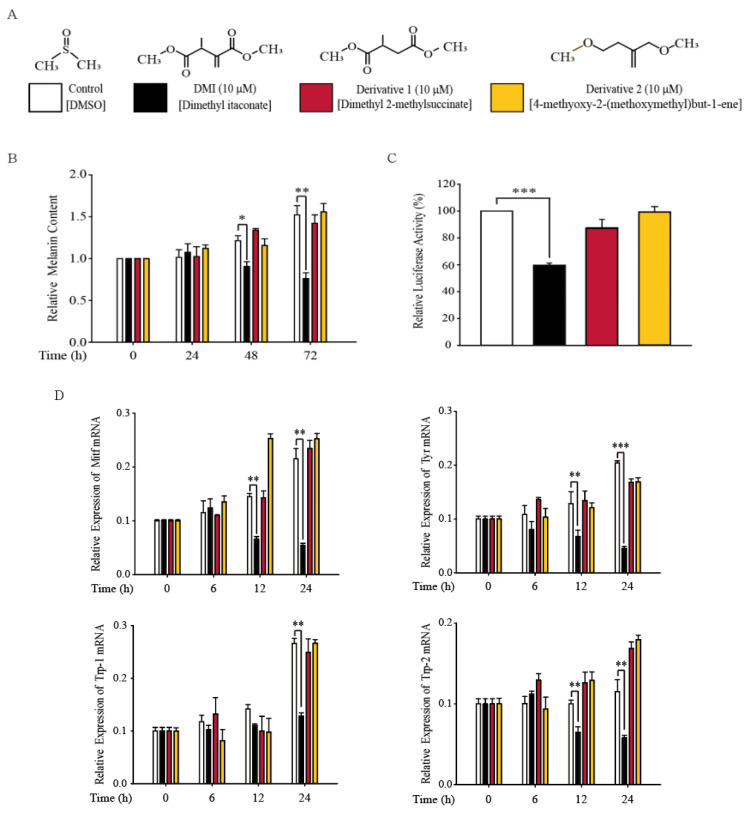
The α,β-unsaturated moiety in DMI is required to inhibit melanogenesis in B16F10 cells. (**A**) Chemical structure of Control (Dimethyl sulfoxide, DMSO), DMI, derivative 1, and derivative 2. (**B**) Derivatives 1 and 2 fail to inhibit the production of melanin in B16F10 cells. B16F10 cells (1 × 10^5^ cells) were exposed to DMI, derivative 1, and derivative 2 at various times, and the production of melanin was measured (*n* = 3). (**C**) Derivatives 1 and 2 fail to inhibit MITF-dependent luciferase activity in B16F10-E-box/M-box-GFP-luciferase cells. Established B16F10-E-box/M-box-GFP-luciferase cells (1 × 10^5^ cells) were exposed to DMI, derivative 1, and derivative 2 for 24 h, and the resulting luciferase activity was measured (*n* = 5). (**D**) Derivatives 1 and 2 fail to inhibit the transcription of MITF and MITF-dependent genes in B16F10 cells. B16F10 cells (1.2 × 10^6^ cells) were exposed to DMI, derivative 1, and derivative 2 at various times, and the levels of Mitf, Tyr, Trp-1, and Trp-2 mRNAs were measured by real-time RT-PCR analysis (*n* = 5).

**Figure 6 antioxidants-12-00692-f006:**
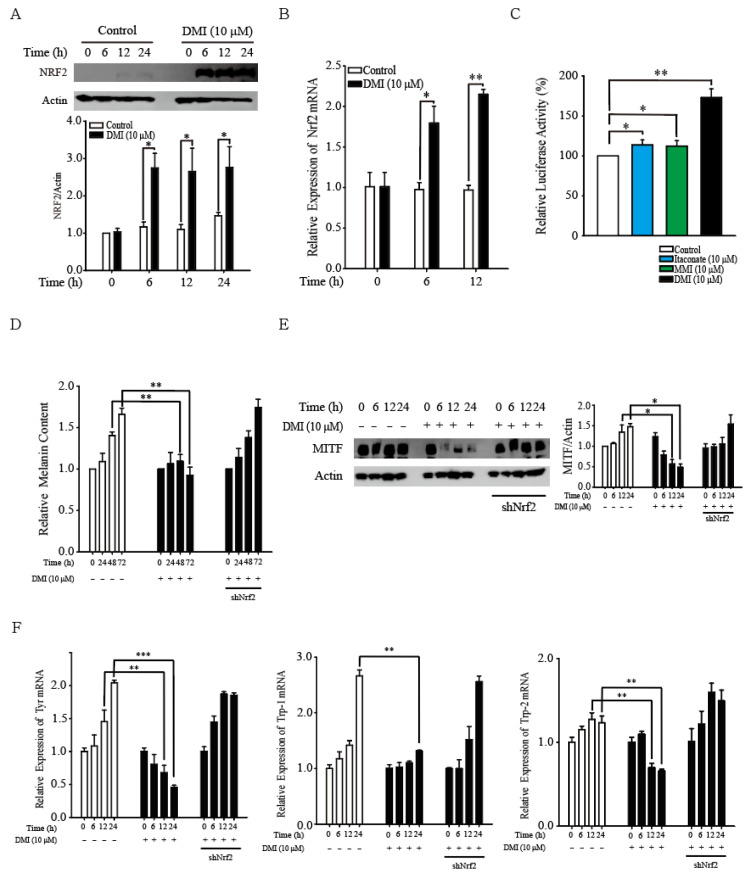
Inhibition of melanogenesis by DMI is dependent on NRF2 in B16F10 cells. (**A**) DMI induces NRF2 in B16F10 cells. B16F10 cells (1.2 × 10^6^ cells) were exposed to DMI at various times, and the protein expression of NRF2 was measured by Western blot analysis. The images of the films were analyzed by densitometry and calculated (*n* = 3). Representative images of the films are provided. (**B**) DMI upregulates the level of Nrf2 mRNAs in B16F10 cells. B16F10 cells (1.2 × 10^6^ cells) were exposed to DMI at various times, and the level of Nrf2 mRNA was measured by real-time RT-PCR analysis (*n* = 5). (**C**) DMI increased ARE-dependent luciferase activity. Established B16F10-ARE-GFP-luciferase cells (1 × 10^5^ cells) were exposed to DMI for 24 h, and the luciferase activity was measured (*n* = 5). (**D**) DMI fails to suppress the production of melanin in B16F10 cells when Nrf2 is silenced. After Nrf2 was knocked down, B16F10 cells (1 × 10^5^ cells) were exposed to DMI, and the production of melanin was measured (*n* = 3). (**E**) DMI fails to inhibit MITF in B16F10 cells when Nrf2 is silenced. After Nrf2 was knocked down, B16F10 cells (1.2 × 10^6^ cells) were exposed to DMI, and the expression of MITF was measured by Western blot analysis. The images of the films were analyzed by densitometry and calculated (*n* = 3). Representative image of the films is provided. (**F**) DMI fails to inhibit the level of Tyr, Trp-1, and Trp-2 mRNAs in B16F10 cells when Nrf2 is silenced. B16F10 cells (1.2 × 10^6^ cells) were exposed to DMI at various times, and the level of Tyr, Trp-1, and Trp-2 mRNAs was measured by real-time RT-PCR analysis (*n* = 5).

**Table 1 antioxidants-12-00692-t001:** The sequence of real-time RT-PCR primers.

	Accession No.	Gene	Primer Sequence
Mouse	NM_001289726.1	Gapdh	Forward: 5′-GGAGAGTGTTTCCTCGTCCC-3′ Reverse: 5′-ACTGTGCCGTTGAATTTGCC-3′
	XM_036165907.1	Mitf	Forward: 5′-AGCGTGTATTTTCCCCACAG-3′ Reverse: 5′-TAGCTCCTTAATGCGGTCGT-3′
	NM_011661.5	Tyr	Forward: 5′-GTCCACTCACAGGGATAGCAG-3′ Reverse: 5′-AGGTGCATTGGCTTCTGGGTA-3′
	XM_006537781.2	Trp-1	Forward: 5′-GGACAGGAAAGCTTTGGGGA-3′ Reverse: 5′-GTCCTCCCGTTCCATTCAGG-3′
	NM_010024.3	Trp-2	Forward: 5′-TACGTGATCACCACGCAACA-3′ Reverse: 5′-ACGTCACACTCGTTCTTCCC-3′
	NM_008777.3	Pah	Forward: 5′-GGGAACGGTGTTCAGGAC-3′ Reverse: 5′-GACAAGAGCCCAGCACCA-3′
	NM_008513.3	Lrp5	Forward: 5′-ACACTC TCTGGGGACACAC-3′ Reverse: 5′-CCTCCAGGGGATCGTAGT-3′
	NM_001122733.1	c-Kit	Forward: 5′-TGTGAACCAACTTCGCCTGA-3′ Reverse: 5′-GCCTGGATTT GCTCTTTGTTG-3′
	NM_008559.3	Mc1r	Forward: 5′-TTCTAGCCATGCTGGCAC-3′ Reverse: 5′-CTGGCTGCGGAAAGCATA-3′
	NM_010902.5	Nrf2	Forward: 5′-CACAGTGCTCCTATGCGTG-3′ Reverse: 5′-TCTGGGCGGCGACTTTATT-3′

**Table 2 antioxidants-12-00692-t002:** Design of oligonucleotides ligated to pLKO.1 vector targeting Mitf and Nrf2.

Gene	Primer Sequence (5′→3′)
shMitf	**Forward:** CCGGATGCTGGAAATGCTAGAATACCTCGAGGTATTCTAGCATTTCCAGCATTTTTTG Age I 21bp sense Loop 21bp antisense Terminal **Reverse:** AATTCAAAAAATGCTGGAAATGCTAGAATACCTCGAGGTATTCTAGCATTTCCAGCAT EcoRI Terminal 21bp sense Loop 21bp antisense
shNrf2	**Forward:** CCGGCCCGAATTACAGTGTCTTAATCTCGAGATTAAGACACTGTAATTCGGGTTTTTG Age I 21bp sense Loop 21bp antisense Terminal **Reverse:** AATTCAAAAACCCGAATTACAGTGTCTTAATCTCGAGATTAAGACACTGTAATTCGGG EcoRI Terminal 21bp sense Loop 21bp antisense

## Data Availability

The data presented in this study are available in the article and Appendix A.

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
