# Peer review of "Dimethyl Itaconate Inhibits Melanogenesis in B16F10 Cells"

_antioxidants, 2023, doi:10.3390/antiox12030692_

Round 1
Reviewer 1 Report (Previous Reviewer 1)
In the Manuscript “Dimethyl itaconate inhibits melanogenesis in B16F10 cells” the Authors indicated the inhibitory effect of dimethyl itaconate (DMI) on melanin synthesis in B16F10 cells, inhibits MITF and downregulates the expression of TYR, TRP1, and TRP2, also in B16F10 cells. They analysed the impact of DMI on MC1R and α-MSH, and ERK1/2 level and the results indicated, that DMI does not inhibit melanogenesis in B16F10 cells lack of Nrf2 expression. Moreover, SAR analysis revealed that the α,β-unsaturated carbonyl moiety in DMI is necessary to inhibit melanogenesis.
MAJOR comments:
1. Most studies were conducted on mouse melanoma cells B16F10. The study would be more valuable if the results were confirmed on human cells (melanoma or normal human melanocytes).
2. The Discussion is not proper. The Authors described their results, referring to the figures. The Discussion must be rewritten – the Authors should compare the results obtained with other publications.
Author Response
Please see in the attachment. Thank you.

Reviewer 2 Report (New Reviewer)
The present manuscript demonstrates the anti-melanogenesis effects of dimethyl itaconate in B16F10 cells and the SAR analysis of itaconic acid. In addition, the link between Nrf2 and melanogenesis is discussed. The manuscript is highly intriguing. However, it has to address various difficulties. please address the point demonstrated in the attached file.

Author Response
Please see in the attachment. Thank you.

Reviewer 3 Report (New Reviewer)
The reviewed article presents the influence of dimethyl itaconate on melanogenesis. The study showed that DMI inhibited melanin synthesis in selected melanoma cell lines. Although the results seem interesting, several issues need to be clarified or improved before deciding to publish the material.
1. Please describe the role of melanin in more detail (concerning protection against UVA, oxidative stress or exogenous substances) in the introduction section.
2. What is the reason for the increase in melanin content in the control samples over time? (this issue also applies to other analyses)
3. Please interpret the increase in cell survival over time (over 100%). Please compare the results presented in Figure 1c with the results shown in Figure 1d.
4. The Authors should explain why NRF2 was required to inhibit melanogenesis by DMI.
Author Response
Please see in the attachment. Thank you.

Reviewer 4 Report (New Reviewer)
The paper is important and well-written, however a couple of melanogenesis measuremen methods should be up-to-date.
line 58, DMEM medium has been mentioned as the medium for investigated melanoma cells. Please mind that high content of tyrosine and DOPA in the medium may additionally interfere with melanin synthesis independently of being a substrate. Suggestion that it may work in a parahormonal way. The best control is maintained in Ham's-F10 medium. See: DOI 10.1111/j.1755-148X.2011.00898.x
line 75 and furthet - the method based on the NaOH extraction of melanoma cells is non-specific. This method may be supplemented with a more specific one, e.g. EPR, and supported with a transient electron microscopy.
Line 148 - what is the proof that the cell number distribution is normal and may be evaluated using the parametric Student's t-test ?
Figure 2A - the figure suggests that DOPA as an intermediate towards DOPAquinone is produced by tyrosinase, as an independent product. Meanwhile, it has been proven that DOPA is a substrate, but it is produced non-enzymatically in the redox exchange reaction. It retardates the initial phase of melanin synthesis starting from L-tyrosine. DOPA may be a substrat for catecholamine reaction to produce dopaquinone, but in the initia, cresolase activity of tyrosinase, it produces dopaquinone directly from tyrosine. Please see e.g. Schallreuter et al. Exp Dermatol, DOI 10.1111/j.1600-0625.2007.00675.x Please also mark the role of PAH (see line 310) in the initiation of melanogenesis (Schallreuter et al, op. cit.)
Author Response
Please see in the attachment. Thank you.

Round 2
Reviewer 1 Report (Previous Reviewer 1)
Accept
Author Response
We thank this reviewer for the positive decision of our manuscript.
Reviewer 2 Report (New Reviewer)
The authors provide satisfactory responses to all queries. But still, need to provide a conclusion section. The conclusion section is missing in the present manuscript.
Author Response
The authors provide satisfactory responses to all queries. But still, need to provide a conclusion section. The conclusion section is missing in the present manuscript.
Response: Complying with the advice of this reviewer, we have supplemented the conclusion in the revised manuscript.
Reviewer 3 Report (New Reviewer)
The manuscript has been improved. The Authors have answered the questions and remarks. However, before publishing, I suggest considering the following suggestions:
1. Figure 1D shows graphs presenting the change in cell number in time. In turn, the caption includes information about cell viability. In my opinion, the Authors should introduce additional graphs showing viability or the number of dead cells.
2. Figure 1 presents results for relative melanin content. Please explain how exactly the melanin content was standardized. Were the results calculated using cell number or total protein content?
3. Please introduce to the article the answer about the reason for the increase in melanin content in the control samples over time.
Author Response
The manuscript has been improved. The Authors have answered the questions and remarks. However, before publishing, I suggest considering the following suggestions:
1. Figure 1D shows graphs presenting the change in cell number in time. In turn, the caption includes information about cell viability. In my opinion, the Authors should introduce additional graphs showing viability or the number of dead cells.
Response: We thank this reviewer for this opinion. It might seem that presenting the results in terms of the viability in the figure legend (Figure 1) might have confused this reviewer in such a way that there exist some cells undergoing cell death. In fact, we seeded cells in the culture dish around 70% confluent and, therefore, cells did not die during our experiment, making impossible for us to plot the number of dead cells.
2. Figure 1 presents results for relative melanin content. Please explain how exactly the melanin content was standardized. Were the results calculated using cell number or total protein content?
Response: We have standardized the melanin based on the cell number.
3. Please introduce to the article the answer about the reason for the increase in melanin content in the control samples over time.
Response: It is highly intuitive to assume that the amount of melanin will increase (Figure 1C and 1D) as cells proliferate (Figure 1D). To this end, it is difficult for us to find out any scientific article(s) pointing out this. We hope that this reviewer understands us on this issue.
Reviewer 4 Report (New Reviewer)
The authors have explained in details their attitude to my remarks. The improved manuscript is well enough to be published in the present form.
Author Response
The authors have explained in details their attitude to my remarks. The improved manuscript is well enough to be published in the present form.
Response: We thank this reviewer for the positive decision of our manuscript.
This manuscript is a resubmission of an earlier submission. The following is a list of the peer review reports and author responses from that submission.
Round 1
Reviewer 1 Report
In the study entitled “Dimethyl itaconate inhibits melanogenesis in B16F10 cells”, the Authors revealed that dimethyl itaconate (DMI) inhibits melanogenesis in B16F10 cells, inhibits MITF and downregulates the expression of key melanogenesis proteins: TYR, TRP1, and TRP2 in B16F10 cells. They also analysed the impact of DMI on MC1R and α-MSH, and ERK1/2 level. They also indicated that the α,β-unsaturated carbonyl moiety in DMI is necessary to inhibit melanogenesis, as well as that DMI does not inhibit melanogenesis in B16F10 cells lack of Nrf2 expression.
Study includes interesting results and it is generally well designed. However, the study has some shortcomings that must to be addressed prior publication:
1. The Authors measured the content of melanin in culture medium. There is a method (widely described in articles) that enables measuring intracellular melanin (in cell pellets). The Authors must measure extracellular melanin level.
2. The Authors should justify your choice of the mouse melanoma cells B16F10 as an experimental model. Studies on human melanocytes (normal and cancer) are recommended.
3. Analysis of melanin level: Was the cell culture medium replaced after 24/48/72 h? This should be specified in the methodology.
4. Statistics – there is a lack of information which post-hoc test was used. Moreover, in many analysis, the Authors compared two means (e.g. Fig.2C) and in my opinion, in such cases t-test should be used rather than ANOVA. Manuscript requires consultation of the exert in the field of statistic. The statistics section must be rewritten.
5. According to the figure 1B, also Monomethyl Itaconate decreased melanin production. Why the Authors choose DMI for the analysis?
6. Western blot analysis: beta-actin blots are not perfectly equal in all samples (Figure 3C). This is understandable, of course, but in this case a densitometric analysis should be performed.
7. The Discussion is poor. It must be extended.
8. The literature has not been prepared in accordance with the journal's editing requirements.
Reviewer 2 Report
In this manuscript, the authors found that DMI inhibits melanogenesis in B16F10 cells and the MC1R-ERK1/2-MITF axis regulated by KEAP1-NRF2 pathway is a molecular target for DMI to inhibit melanogenesis in B16F10 cells. This study focused on the α, β-unsaturated carbonyl moiety in DMI is required to suppress melanogenesis in B16F10 cells. However, results were too prelimilary to publish.
1. According to Figure 1D, Dimethyl Itaconate (DMI) does not inhibit the proliferation of melanoma cells, indicating that it cannot play a role in treating tumors. Why should the author choose B16F10 instead of melanocytes (such as HEM cells) for research?
2. The authors should use MTT/MTS or CCK8 to confirm that DMI does not inhibit cell proliferation.
3. In the section of "2.2 Measurement of Melanin", the author should cite references to explain why the melanin content in the culture medium, rather than in cells, is detected.
4. In Figure 2B, Figure 3C and Figure 4D, what time point does "0, 6, 12, 24h" of the control group represent?
5. The authors should cite all papers you use properly. For example, lines 171-174, 193-195, and 213-217 are missing citations.
6. In line 271, "These results illustrate that NRF2 activation by DMI ·····", but in Figure 6, there is no direct result indicating that DMI activates NRF2. In addition, in line 314-316, " we propose that the identification of new NRF2 activators can serve as a feasible strategy for development of new anti-melanogenic agents". The authors should give direct evidence whether DMI can act as an activator of NRF2.
Reviewer 3 Report
In the present manuscript Bo-Yeong Yu et al investigate dimethyl itaconate (DMI) effects in melanogenesis. They observed that dimethyl itaconate inhibits melanogenesis in B16F10 cells. DMI inhibits microphthalmia-associated transcription factor (MITF) and downregulates the expression of some MITF target genes. The present study reveals the MC1R-ERK1/2-MITF axis regulated by the KEAP1-NRF2 pathway is the molecular target responsible for the inhibition of melanogenesis by DMI in B16F10 cells.
The results obtained in this study have a promise for future investigation of melanogenesis and for development of new anti-melanogenic agents. As such, it is suitable for publication in Antioxidants in its present form, however, prior to acceptance, the manuscript needs some minor changes in order to improve the quality of the manuscript.
In the materials and methods section the authors should to include if the authors verify the identity and purity of B16F10 mouse cell line. Routine testing of cell cultures is critical as cell lines frequently undergo misidentification, cross-contamination, and genetic drift.
The authors do not mention the number of different preparations used to make western blots analysis. And they should performed densitometric scanning of blots in each condition and show the resulting histograms.
It would be advisable not to put the same western blot analysis on two different figures (figure 3C and Supplementary figure 2). The authors should change one of them.
Supplementary figure 2 is not well aligned with the text.
This manuscript is technically well performed and the results and conclusions are sound and novel. In my opinion this manuscript is suitable for publication in Antioxidants with minor changes.